



**Validation and application of optimal ionospheric shell height model**
**for single-site TEC estimation**
Jiaqi Zhao[1], Chen Zhou[1]
School of Electronic Information, Wuhan University, Wuhan, 430072, China
Corresponding to: chenzhou@whu.edu.cn
**Abstract**
We recently proposed a method to establish optimal ionospheric shell height model
based on the international GNSS service (IGS) station data and the differential code
bias (DCB) provided by Center for Orbit Determination in Europe (CODE) during the
time from 2003 to 2013. This method is very promising for DCB and accurate total
electron content (TEC) estimation by comparing to traditional fixed shell height method.
However, this method is basically feasible only for IGS stations. In this study, we
investigate how to apply the optimal ionospheric shell height derived from IGS station
to non-IGS stations or isolated GNSS receivers. The intuitional and practical method to
estimate TEC of non-IGS stations is based on optimal ionospheric shell height derived
from nearby IGS stations. To validate this method, we selected two dense networks of
IGS stations located in US and Europe region. Two optimal ionospheric shell height
models are established by two reference stations, namely GOLD and PTBB, which are
located at the approximate center of two selected regions. The predicted daily optimal
ionospheric shell heights by the two models are applied to other IGS stations around

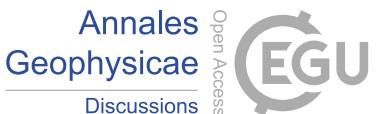

these two reference stations. Daily DCBs are calculated according to these two optimal
shell heights and compared to respective DCBs released by CODE. The validation
results of this method present that 1) Optimal ionospheric shell height calculated by
IGS stations can be applied to its nearby non-IGS stations or isolated GNSS receivers
for accurate TEC estimation. 2) As the distance away from the reference IGS station
becomes larger, the DCB estimation error becomes larger. The relation between the
DCB estimation error and the distance is generally linear.

**Keyword**
Ionospheric shell height, Single layer model (SLM), Differential code bias (DCB), Total
electron content (TEC)

**Introduction**
Dual-frequency GPS signals propagation are affected effectively by ionospheric
dispersive characteristic. While, by taking advantage of this property, ionospheric TEC
along the path of signal can be estimated by using differencing the pseudorange or
carrier phase observations from dual-frequency GPS signals. Carrier phase
leveling/smoothing of code measurement is widely adopted to improve the precision of
absolute TEC observations (Mannucci et al. 1998; Horvath and Crozier 2007). In
general, it is considered that the derived TEC in carrier phase leveling/smoothing



technique consists of slant TEC (STEC), the combination differential code bias (DCB)
of satellite and receiver, multipath effects and noise. The DCB is usually considered as
the main error source and could be as large as several TECu (Lanyi and Roth 1988;
Warnant 1997).

For TEC and DCB estimations, mapping function with single layer model (SLM)

assumption have been intensively studied for many years. Sovers and Fanselow (1987)
firstly simplified the ionosphere to a spherical shell. They set the bottom and the top
side of the ionospheric shell as h-35 and h+75 km, where h is taken to be 350 km above
the surface of the earth and allowed to be adjusted. In this model, the electron density
was evenly distributed in the vertical direction. Based on this model, Sardón et al. (1994)
introduced the Kalman filter method for real-time ionospheric VTEC estimation.
Klobuchar (1987) assumed that STEC equals VTEC multiplied by the approximation
of the standard geometric mapping function at the mean vertical height of 350 km along
the path of STEC. Lanyi and Roth (1988) further developed this model into a single
thin-layer model, and proposed the standard geometric mapping function and the
polynomial model. The single thin-layer model assumed that the ionosphere is
simplified by a spherical thin shell with infinitesimal thickness. Clynch et al (1989)
proposed a mapping function in the form of a polynomial by assuming a homogeneous
electron density shell between altitudes of 200 and 600 km. Mannucci et al (1998)
presented an elevation scaling mapping function derived from extended slab mode.
There are also many modified mapping function according to the standard geometric



mapping function. Schaer (1999) proposed the modified standard mapping function
using a reduced zenith angle. Rideout and Coster (2006) presented a new mapping
function which replaces the influence of the shell height by an adjustment parameter,
and set the shell height as 450 km. Smith et al (2008) modified the standard mapping
function by using a complex factor. Based on the electron density field derived from
the international reference ionosphere (IRI), Zus et al (2017) recently developed an
ionospheric mapping function at fixed height of 450 km with dependence on time,
location, azimuth angle, elevation angle, and different frequencies.

Ionospheric shell height is considered to be the most important parameter for

mapping function, and the shell height is typically set to a fixed value between 350 and
450 km (Lanyi and Roth 1988; Mannucci et al. 1998). Birch et al. (2002) proposed an
inverse method for estimate the shell height by using simultaneous VTEC and STEC
observations, and suggested the shell height is preferred to be a value between 600 and
1200 km. Nava et al. (2007) presented a shell height estimation method by minimizing
the mapping function errors, this method is referred as the "coinciding pierce point"
technique. Their results indicated that the suitable shell heights for the mid-latitude is
400 km and 500 km during the geomagnetic undisturbed conditions and disturbed
conditions, respectively. In the case of the low-latitude, the shell height at about 400
km is suitable for both quiet and disturbed geomagnetic conditions. Jiang et al. (2018)
applied this technique to estimate the optimal shell height for different latitude bands.
In their case, the optimal layer height is about 350 km for the entire globe. Brunini et



al. (2011) studied the influence of the shell height by using an empirical model of the
ionosphere, and pointed out that a unique shell height for whole region does not exist.
Li et al. (2017) applied a new determination method of the shell height based on the
combined IGS GIMs and the two methods mentioned above to the Chinese region, and
indicated that the optimal shell height in China ranges from 450 to 550 km. Wang et al.
(2016) studied the shell height for grid-based algorithm by analyzing goodness of fit
for STEC. Lu et al. (2017) applied this method to different VTEC models, and
investigated the optimal shell heights at solar maximum and at solar minimum.

In the recent study by Zhao and Zhou (2018), a method to establish optimal

ionospheric shell height model for single station VTEC estimation has been proposed.
This method calculates the optimal ionospheric shell height with regards to minimize
$|\Delta DCB|$ by comparing to the DCB released by CODE. Five optimal ionospheric shell
height models were established by the proposed method based on the data of five IGS
stations at different latitudes and the corresponding DCBs provided by CODE during
the time 2003 to 2013. For the five selected IGS stations, the results have shown that
the optimal ionospheric shell height models improve the accuracies of DCB and TEC
estimation comparing to fixed ionospheric shell height of 400 km in a statistical sense.
We also found that the optimal ionospheric shell height show 11-year and 1-year
periods and is related to the solar activity, which indicated the connection of the optimal
shell height with ionospheric physics.

While the proposed optimal ionospheric shell height model is promising for DCB



and TEC estimation, this method cannot be implemented to isolated GNSS receivers
not belonging to IGS stations. The purpose of this study is to investigate the application
of the optimal ionospheric shell height derived from IGS station to non-IGS stations.
By considering the spatial correlation of ionospheric electron density, it is intuitional
and practical to adopt the optimal ionospheric shell height of a nearby IGS station for
the non-IGS stations.

The purpose of this study is to investigate the feasibility of applying the optimal

ionospheric shell height derived from IGS station to nearby non-IGS GNSS receivers
for accurate TEC/DCB estimation. By selecting two different regions in U.S. and
Europe with dense IGS stations, we calculate the daily DCBs of 2014 by using the
optimal ionospheric shell heights derived from 2003-2013 data of two central stations
in two regions. We also try to find the DCB estimation error and its relation to distance
away from the central reference station.

**Method**
In (Zhao and Zhou, 2018), we proposed a concept of optimal ionospheric shell height
for accurate TEC and DCB estimation. Based on daily data of single site, this approach
searches daily optimal ionospheric shell height, which minimizes the difference
between the DCBs calculated by VTEC model for single site and reference values of
DCB. For a single site, its long-term daily optimal ionospheric shell heights can be
estimated and then modeled. In our case, the polynomial model (Lanyi and Roth 1988;





Wild 1994) is applied to estimate satellite and receiver DCBs, and the DCBs provided
by CODE are used as the reference.

In the polynomial model, the VTEC is considered as a Taylor series expansion in

latitude and solar hour angle, which is expressed as follows:
$$T_V(\varphi, S) = \sum_{i=0}^{m} \sum_{j=0}^{n} E_{ij} (\varphi - \varphi_0)^i (S - S_0)^j \qquad (1)$$

where $T_V$ denotes VTEC. $\varphi$ and $S$ denote the geographic latitude and the solar
hour angle of IPP, respectively; $\varphi_0$ and $S_0$ denote $\varphi$ and $S$ at regional center.
$E_{ij}$ is the model coefficient. $m$ and $n$ denote the orders of the model. A polynomial
model fits the VTEC over a period of time. In our case, 8 VTEC models are applied per
day, and DCB is considered as constant in one day. Since our analysis is based on long-
term single site data, we set $m$ and $n$ to 4 and 3, respectively. Huang and Yuan (2014)
applied the polynomial model with the same orders to TEC estimation.

Based on the thin shell approximation, the observation equation can be written as:

$$T_{os}^{PRN}(\varphi, S) = T_V(\varphi, S) \cdot f(z) + DCB^{PRN} \qquad (2)$$

where $T_{os}^{PRN}$ is slant TEC calculated by carrier phase smoothing, the superscript $PRN$
denotes GPS satellite. $DCB^{PRN}$ denotes the combination of GPS satellite and receiver
DCB. $z$ denotes the zenith angle of IPP. According to Lanyi and Roth (1988), the
standard geometric mapping function $f(z)$ is expressed as follows:
$$f(z) = 1 / \cos(z) \qquad (3)$$

$$z = \arcsin \frac{\text{Re} \cdot \cos El}{\text{Re} + h} \qquad (4)$$




where  Re  denotes the earth's radius,  $El$  denotes the elevation angle, and $h$ denotes
the thin ionospheric shell height. Note that $h$ also affects the location of IPP.
To estimate DCBs, The method above requires a definite thin shell height value.
Conversely, if we get the daily solutions of DCBs, the optimal ionospheric shell height
can be estimated. The optimal ionospheric shell height is assumed to be between 100
and 1000 km and is defined as the shell height with the minimum difference between
$DCB^{PRN}$ and the reference values. This optimization problem can be written as:
$$\min_{100<h<1000} mean\left(\left|\mathbf{DCB_{ref}} - \mathbf{DCB}\right|\right) \text{ s.t. } \mathbf{T} = \mathbf{\Phi}\cdot\mathbf{E} + \mathbf{\theta}\cdot\mathbf{DCB} \qquad (5)$$

where  $h$  is the daily optimal ionospheric shell height,  $\mathbf{DCB_{ref}}$  denotes the vector of
the reference values of DCBs,  s.t.  is the abbreviation for subject to,
$\mathbf{T} = \mathbf{\Phi}\cdot\mathbf{E} + \mathbf{\theta}\cdot\mathbf{DCB}$  is the matrix form of all the observation equations in one day,  $\mathbf{T}$
denotes the vector of $T_{os}$,  $\mathbf{E}$  corresponds to the coefficients of the models,  $\mathbf{DCB}$  is
the vector of $DCB^{PRN}$,  $\mathbf{\Phi}$  and  $\mathbf{\theta}$  are the coefficient matrix of  $\mathbf{E}$  and  $\mathbf{DCB}$,
respectively.
After the method above is applied to 11-year data, the estimated optimal
ionospheric shell heights can be modeled by a Fourier series, which is expressed as
follows:
$$h(x) = a_0 + \sum_{n=1}^{k}\left(a_n \cos\frac{2n\pi x}{L} + b_n \sin\frac{2n\pi x}{L}\right) \qquad (6)$$

where  $k$  is the order of Fourier series and is set to 40,  $a_n$  and  $b_n$  are the model
coefficients,  $x$  is the time, and  $L$  is the time span which equals to 4018 days. The





maximum frequency of model is 40/L≈0.01 per day. By least square method, the model
coefficients can be estimated. The estimated daily optimal ionospheric shell height
$h(x)$ by the model is then applied to other neighboring stations in this region. By using
$h(x)$, we can validate the TEC and DCB estimation.

**Experiment and Results**
The previous section introduced a method to establish daily optimal ionospheric shell
height model based on single site with reference values of DCBs. To analyze the
improvement of DCB estimation by this model for the reference station and other
neighboring stations, we present two experiments to evaluate and validate this method
by using IGS stations located in U.S. and Europe region. To ensure the accuracy and
consistency of DCB, we only select IGS stations with pseudorange measurements of
P1 code, and whose receiver DCBs have been published by CODE.

Figure 1 presents the location and distribution of the selected IGS stations in two

regions. Table 1 presents the information of the geographical location, distance to
reference station in each region and receiver types of all stations. Based on the RINEX
data of GOLD station in Region I and PTBB station in Region II during the period of
2003-2013, two separate optimal ionospheric shell height models for each region are
established by the aforementioned method. Then the model are applied to DCB
estimation in 2014 for all the other stations in each region. Note that reference GOLD





and PTBB stations are marked with black triangle in the figure. The other neighboring
stations are located in different orientations of GOLD and PTBB with different
distances, which range from 136 to 1159 km for region I and range from 190.82 to
1712.27 km for region II. In the table, the receiver type is corresponding to 2003~2014
for GOLD and PTBB, and 2014 for the other stations. In region I, the receiver type of
GOLD have been changed once in September 2011. The five selected stations used four
receiver types in 2014; TABV and PIE1 had the same receiver type. In region II, there
are nine receiver types for the sixteen stations. The receiver type of PTBB have changed
twice in 2006.



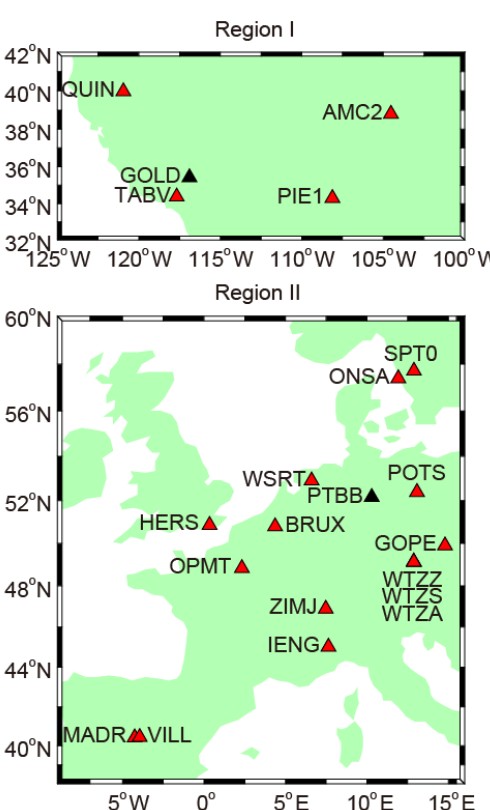


**Fig.1** Geographical location of the selected IGS stations in U.S. region (Region I) and

Europe region (Region II).



**Table 1** Information for the stations

| Name | Latitude (deg) | Longitude (deg) | Distance to GOLD or PTBB (km) | Receiver type |
|------|----------------|-----------------|-------------------------------|---------------|
| GOLD | 35.42 | -116.89 | 0 | ASHTECH Z-XII3 ~ 2011-09-14<br>JPS EGGDT          2011-09-19 ~ |
| TABV | 34.38 | -117.68 | 136.67 | JAVAD TRE_G3TH DELTA |
| QUIN | 39.97 | -120.94 | 619.55 | ASHTECH UZ-12 |





| PIE1 | 34.30 | -108.12 | 810.51 | JAVAD TRE_G3TH DELTA |
|------|-------|---------|--------|----------------------|
| AMC2 | 38.80 | -104.52 | 1159.09 | ASHTECH Z-XII3T |
| PTBB | 52.15 | 10.30 | 0 | SEPT POLARX2 2006-07-25~ 2006-11-13 ASHTECH Z-XII3T    else |
| POTS | 52.38 | 13.07 | 190.82 | JAVAD TRE_G3TH DELTA |
| WSRT | 52.91 | 6.60 | 264.92 | AOA SNR-12 ACT |
| WTZA | 49.14 | 12.88 | 381.28 | ASHTECH Z-XII3T |
| WTZS | 49.14 | 12.88 | 381.28 | SEPT POLARX2 |
| WTZZ | 49.14 | 12.88 | 381.28 | JAVAD TRE_G3TH DELTA |
| GOPE | 49.91 | 14.79 | 401.51 | TPS NETG3 |
| BRUX | 50.80 | 4.36 | 439.03 | SEPT POLARX4TR |
| ONSA | 57.40 | 11.93 | 593.72 | JPS E_GGD |
| ZIMJ | 46.88 | 7.47 | 620.79 | JAVAD TRE_G3TH DELTA |
| SPT0 | 57.72 | 12.89 | 641.78 | JAVAD TRE_G3TH DELTA |
| OPMT | 48.84 | 2.33 | 674.24 | ASHTECH Z-XII3T |
| HERS | 50.87 | 0.34 | 705.38 | SEPT POLARX3ETR |
| IENG | 45.02 | 7.64 | 816.64 | ASHTECH Z-XII3T |
| VILL | 40.44 | -3.95 | 1696.62 | SEPT POLARX4 |
| MADR | 40.43 | -4.25 | 1712.27 | JAVAD TRE_G3TH DELTA |


Figure 2 presents the estimated daily optimal ionospheric shell height of GOLD
and PTBB during the period from 2003 to 2013. The left panel shows the variation of
the daily optimal ionospheric shell height and the fitting result by (6). From the overall
trend, the variations of daily optimal ionospheric shell height for both two stations
appear wave-like oscillation during the 11 years period. In the right panel, the statistical



result are fitted by a normal distribution. The mean and the standard deviation (STD)
of the normal distribution are 714.3 and 185.4 km for GOLD, respectively. The mean
and STD value for PTBB is 416.4 and 184.1 km, respectively.

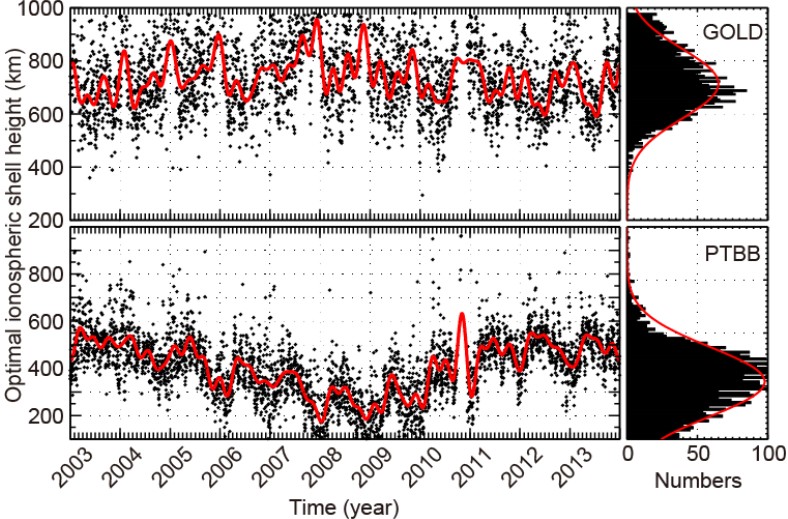


**Fig.2** Variation of the daily optimal ionospheric shell height (black) and the fitting
result (red)

Figure 3 presents the amplitude spectra of the daily optimal ionospheric shell
height of two reference stations estimated by the Lomb-Scargle analysis (Lomb 1976;
Scargle 1982). As can be found in Figure 3, the peaks correspond to 11-year, 1-year, 6-
month and 4-month cycles. The amplitudes of 11-year and 1-year cycles are more
evident than other periods in both two stations. Note that the frequencies above 0.01
per day are discarded because of their small amplitudes. As mentioned earlier, 0.01 per
day is about the maximum frequency of (6). This result shows that the optimal



ionospheric shell height of GOLD and PTBB is periodic, and the 40th-order of Fourier
series is suitable for modelling its variation.

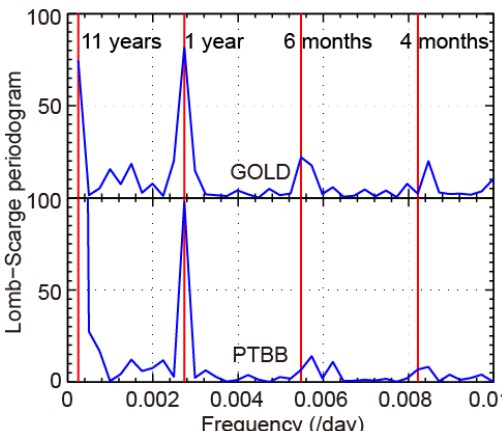

**Fig.3** Lomb-Scargle spectra of the daily optimal ionospheric shell height

We establish two optimal ionospheric shell height models for each region by the

40th-order of Fourier series based on the 11-year data of GOLD and PTBB. To
investigate the availability zone of the optimal ionospheric shell height model, we apply
the model to the stations of each region as shown in Figure 1 and Table 1. Based on the
predicted daily optimal ionospheric shell heights in 2014 calculated by the model of
GOLD and PTBB, the DCBs in all stations of each region are estimated in the form of
single station by the polynomial model mentioned earlier. The difference of DCBs in
all station in each region calculated by the optimal ionospheric shell height model from
each reference station and DCBs provided by CODE is then compared to the difference
of DCBs calculated by fixed ionospheric shell height (400 km) and DCBs released by





CODE.
Figure 4 shows the daily average differences of DCBs calculated by the model and
DCBs of each stations provided by CODE in 2014, and the differences of DCBs
calculated by the fixed ionospheric shell height (400 km) and DCBs released by CODE
in 2014. The panels for the stations are arranged by their distances to reference station,
this is also applied to the following table; from the top panels to the bottom panels, the
distance of the corresponding station to the reference station gradually increases. The
left and right panels show the daily differences and the histograms of the statistical
results in 2014, respectively. For all of the stations, the daily average differences of
DCBs calculated by the optimal ionospheric shell height model are reduced compared
to the fixed ionospheric shell height. For GOLD and TABV, the reductions are
appropriate, the daily average ΔDCBs around 0 have the most days. For the other
stations, the reductions are so much that most of the average ΔDCBs are negative. This
result shows the improvement of the model seems to be related with the distance to
GOLD. Note that some days no result because of missing data. Figure 5 is the same
format as Figure 4, which presents the results of Region II. By comparing to the results
of fixed ionospheric height, Figure 5 also indicates that the ΔDCB of optimal
ionospheric shell heights with PTBB prediction is more concentrated distributed around
0 in a statistical sense. Both Figure 4 and Figure 5 present the accuracy of DCB
estimation by using optimal ionospheric heights from reference station, namely GOLD
and PTBB in this study, can be improved.



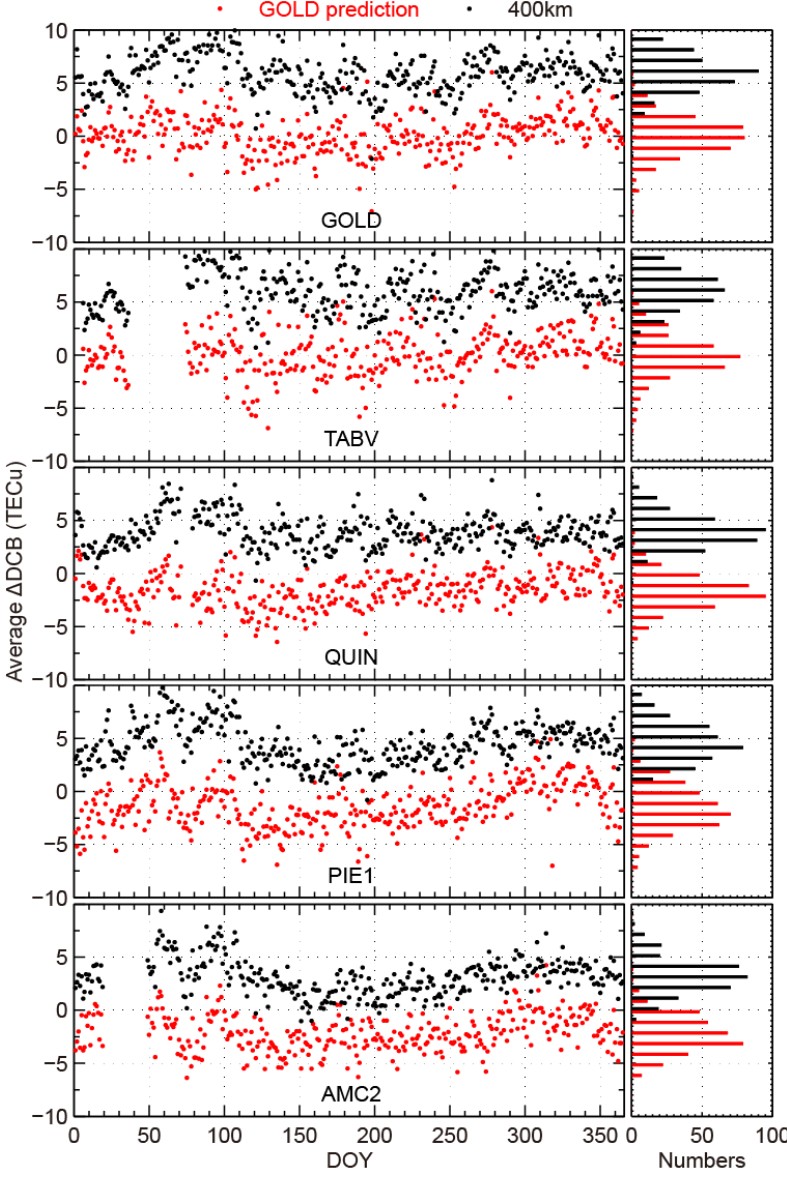


**Fig.4** Comparisons of the average ΔDCB calculated by the predicted optimal
ionospheric shell heights (red dots) and by the fixed ionospheric shell height (black dots)
in 2014 for stations in Region I.




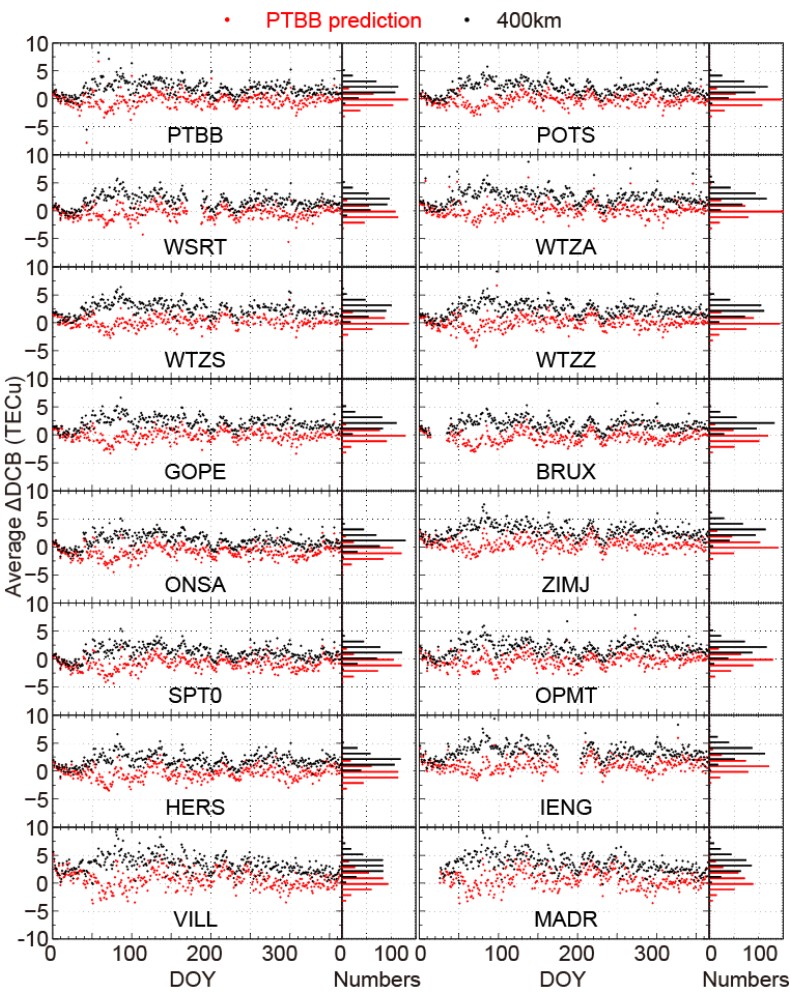


**Fig.5** Comparisons of the average ΔDCB calculated by the predicted optimal

ionospheric shell heights (red dots) and by the fixed ionospheric shell height (black dots)

in 2014 for stations in Region II.


Table 2 presents the quantitative statistical results of average ΔDCB in 2014. For





all the stations in each region, the mean values and the root mean squares (RMS) by the
optimal ionospheric shell height model are smaller than by the fixed ionospheric height.
For Region I, the improvements of TABV are the most significant. Their mean values
are reduced to 0.12 and 0.08 TECu, respectively; the root mean squares are reduced by
4.43 and 4.33 TECu, respectively. For Region II, the improvement for DCB estimation
are the most obvious for WTZZ, with mean value of ΔDCB decreases from 2.34 to 0.02.
We could note that TABV and WTZZ station are quite close to the reference stations in
each region.

**Table 2** Statistical results of mean (ΔDCB) in 2014

| Station | Average ΔDCB (TECu) Optimal Ionospheric Height | | Average ΔDCB (TECu) Fixed Ionospheric Height | |
|---|---|---|---|---|
| | Mean | RMS | Mean | RMS |
| GOLD | 0.12 | 1.82 | 5.96 | 6.25 |
| TABV | 0.08 | 2.04 | 6.06 | 6.37 |
| QUIN | -1.60 | 2.31 | 3.91 | 4.19 |
| PIE1 | -1.38 | 2.50 | 4.46 | 4.84 |
| AMC2 | -2.12 | 2.75 | 3.09 | 3.53 |
| PTBB | -0.28 | 1.23 | 1.82 | 2.26 |
| POTS | -0.27 | 1.00 | 1.84 | 2.18 |
| WSRT | -0.41 | 1.14 | 1.65 | 2.10 |
| WTZA | 0.09 | 1.20 | 2.38 | 2.73 |
| WTZS | 0.14 | 0.99 | 2.48 | 2.76 |
| WTZZ | 0.02 | 1.14 | 2.34 | 2.65 |



| GOPE | -0.17 | 1.00 | 2.12 | 2.41 |
| BRUX | -0.42 | 1.12 | 1.86 | 2.13 |
| ONSA | -0.88 | 1.40 | 1.10 | 1.63 |
| ZIMJ | 0.48 | 1.17 | 2.87 | 3.13 |
| SPT0 | -0.84 | 1.40 | 1.14 | 1.67 |
| OPMT | -0.29 | 1.21 | 1.93 | 2.35 |
| HERS | -0.37 | 1.19 | 1.84 | 2.19 |
| IENG | 1.05 | 1.57 | 3.44 | 3.69 |
| VILL | 0.59 | 1.67 | 3.30 | 3.66 |
| MADR | 0.66 | 1.71 | 3.50 | 3.86 |


Figure 6 and Figure 7 present the relation between the statistical results of average
ΔDCB and the distance to reference stations in each region. The left and the right panels
in each figure show the relation of the absolute mean value and the root mean square
with the distance to GOLD and PTBB, respectively. For all of the stations, the optimal
ionospheric shell height model improves the accuracies of DCB estimation compared
to the fixed ionospheric shell height in a statistical sense; both of the absolute mean
values and the root mean squares become smaller. For the optimal ionospheric shell
height model, the absolute mean values present a positive correlation with the distance
to reference station GOLD and PTBB in each region, as well as the root mean squares.
By using the linear regression, for Region I, the absolute mean value increases at a rate
of about 1.84 TECu per 1000 km and start at about 0.05 TECu. The RMS value
increases at a rate of about 0.75 TECu per 1000 km and starts at about 1.87 TECu.





According to the fitting results, the absolute mean value and the RMS less than 1 TECu
and 2.25 TECu in the region around GOLD with a radius of 500 km, and less than 2
TECu and 2.62 TECu for the region with a radius of 1000 km. For Region II, the
absolute mean value increases at a rate of about 0.30 TECu per 1000 km and start at
about 0.25 TECu. The RMS value increases at a rate of about 0.41 TECu per 1000 km
and starts at about 1.01 TECu. According to the fitting results, the absolute mean value
and the RMS less than about 0.40 TECu and 1.21 TECu in the region around PTBB
with a radius of 500 km, and less than about 0.55 TECu and 1.42 TECu for the region
with a radius of 1000 km. For the two regions, the RMSs presents stronger linear
relation with distance comparing to the means.

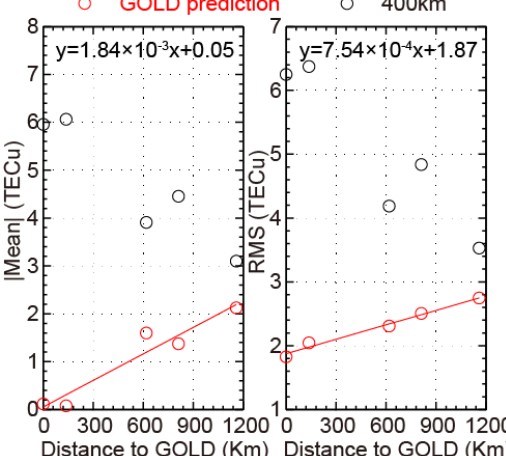


**Fig.6** Relation of the accuracy for DCB estimation with the distance to GOLD. The red
lines are the linear fitting results




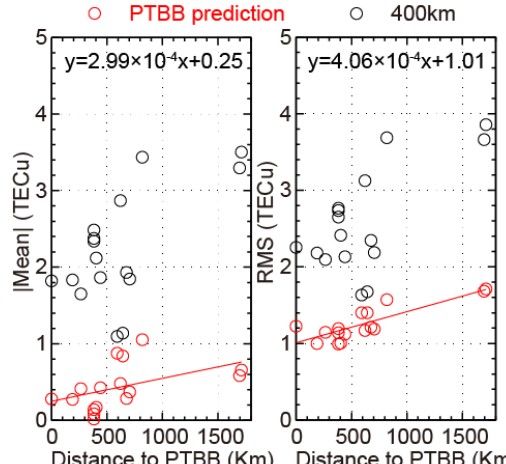


**Fig.7** Relation of the accuracy for DCB estimation with the distance to PTBB. The red

lines are the linear fitting results



**Summary**

In this study, we investigate the implementation and validation of optimal ionospheric

shell height derived from IGS station to non-IGS station or isolated GNSS receiver. We

establish two optimal ionospheric shell height models by the 40th-order of Fourier

series based on the data of IGS station GOLD and PTBB in two separate regions These

two models are applied to the stations in each region, where the distance to GLOD

ranges from 136.67 to 1159.09 km and the distance to PTBB ranges from 190.82 to

1712.27 km. The main findings are summarized as follows:

1) The optimal ionospheric shell height model improves the accuracy of DCB





estimation comparing to the fixed shell height for all of the stations in a statistical
sense. This results indicate the feasibility of applying the optimal ionospheric shell
height derived from IGS station to other neighboring stations. The IGS station can
calculate and predict the daily optimal ionospheric shell height, and then release
this value to the nearby non-IGS stations or isolated GNSS receivers.
2)  For other station in each region, the error of DCB by the optimal ionospheric shell
height increases linearly with the distance to the reference GOLD and PTBB station.
For the mean and the RMS of the daily average ΔDCBs, in region I, the slopes are
about 1.84 and 0.75 TECu per 1000 km; in region II, the slopes are about 0.30 and
0.41 TECu per 1000 km. This results indicate the horizontal spatial correlation of
regional ionospheric electron density distribution. For different region, the error at
0 km (i.e. the error for the reference station) is different, which should be also
considered.
As the requirement of this experiment, we just analyze two regions in mid-latitude
due to the insufficiency of long-term P1 data. We also ignore the orientation of isolated
GPS receivers to the reference station.

**Acknowledgments**
This study is based on data services provided by the IGS (International GNSS Service)
and CODE (the Center for Orbit Determination in Europe). This work is supported by





the National Natural Science Foundation of China (NSFC grant 41574146 and

41774162).

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
