# Peer review of "Validation and application of optimal ionospheric shell height model"

_Annales Geophysicae, 2018_

## Referee Comment (RC1) · Anonymous Referee #1 · 3 Aug 2018

The manuscript is well written and clear. This paper models the optimal thin layer altitude as 40th order Fourier series. The optimal atitude coefficients are estimated at a reference station and used for estimating the thin shell altitude of the near stations. This approach is very useful for precisely estimating the TEC and DCB in not IGS-platform.

It would be interesting to see what are the performances in DCB estimation if the number of coefficients of the Fourier series changes, it would be interesting to see what is the minum number of coefficients required.

There is in addition a typo comment, on page 7 line 132, you should insert the IPP

acronym, that has not been specified before.

---

## Author Comment (AC1) · 30 Aug 2018

We thank referee #1 for careful reading and valuable comments on the manuscript. Accordingly, we have modified the text. All the modifications and changes are shown in the revised manuscript "manuscript-version2" in red font. Our responses to the referee's comments are listed below and the file "Response to RC1". The two files are in supplement.

Response to Referee #1

COMMENTS TO THE AUTHOR: Referee #1: Interactive comment on "Validation and

application of optimal ionospheric shell height model for single-site TEC estimation" by Jiaqi Zhao and Chen Zhou

Comments:

1.The manuscript is well written and clear. This paper models the optimal thin layer altitude as 40th order Fourier series. The optimal altitude coefficients are estimated at a reference station and used for estimating the thin shell altitude of the near stations. This approach is very useful for precisely estimating the TEC and DCB in not IGS platform.

Reply:

We thank the referee for the encouraging evaluation on our study, which has driven us into a deeper investigation.

2.It would be interesting to see what are the performances in DCB estimation if the number of coefficients of the Fourier series changes, it would be interesting to see what is the minim number of coefficients required.

Reply:

We thank the referee for this valuable comment. In our study, the order of Fourier series is preliminarily set to 40. For one station, the outstanding frequencies of optimal thin layer altitude are only a few, it is possible to reduce the number of coefficients. Maybe less coefficient, clearer physical relationship.

3.There is in addition a typo comment, on page 7 line 132, you should insert the IPP acronym that has not been specified before.

Reply:

We thanks the referee for careful reading and pointing out this mistake. We have corrected it. Please see page 7 line 135 in the revised manuscript.

Please also note the supplement to this comment:
https://www.ann-geophys-discuss.net/angeo-2018-73/angeo-2018-73-AC1-
supplement.zip
* * *
Interactive
comment

---

## Referee Comment (RC2) · Anonymous Referee #2 · 3 Sep 2018

**1   General comments**

This article presents a novel approach to estimate GPS permanent stations DCBs in a radius less than 2000 km from a mid-latitude IGS station. It is a validation of a technique recently developed by the same authors. An optimum ionospheric shell height is estimated using the assumption that the IGS DCBs represent reliable values. This study covers a complete solar cycle for the estimation of the ionospheric shell height at a reference station and one year for the tests with additional stations. I think that the manuscript in its present form lacks of necessary discussion on the limitations of the assumptions made in this work and that a number of points need to be explained

deeper. I therefore suggest major revisions.

**2 Specific comments**

The ionospheric height for the reference station GOLD appear to be very hight: in average it is 712 km. By an ionospheric point of view, the shell height should correspond to the height of the ionosphere barycentre, i.e. higher than hmF2 of about 100-150 km. There is a long-lasting debate on the operational shell height to use for the thin shell approximation of the ionosphere and the authors recall many of the publications discussing this problem. While it is true that some authors allow altitudes as high as 1200 km, care should be taken to understand if the obtained shell height are reliable. In this work an elevation mask of $15°$ has been used. Under many conditions this elevation mask could be too low and introduce a large uncertainty on the optimum shell height (see for instance the discussions of Rama Rao et al. 2006, recall that under some conditions they even obtained unphysical negative shell height).

It is not clear why the technique proposed for ionospheric shell height estimation cannot be implemented to isolated GNSS receivers not belonging to IGS stations (line 107).

A Fourier model of the shell height is constructed for GOLD and PTBB for a complete solar cycle between 2003 and 2013. This model does not include any input regarding solar activity. It is well known that the current solar cycle is considerably less strong than the previous one. The ionosphere development has also been substantially lower. Thus it is also expected the the optimum shell height should follow a different pattern. A discussion on this point is essential for the correct understanding of this work.

line 52: the work of Sardón et al. (1994) was not oriented towards real-time ionospheric VTEC, but to develop a technique of prediction of DCBs under adverse conditions (antispoofing, ionospheric disturbances).

line 77: specify that the Nava et al. (2007) technique uses multiple stations to obtain a "coinciding pierce point".

line 125-126: the polynomial model is referred to Lanyi and Roth (1988).However the expression used in this article does not correspond to the one used by those authors.

Line 132: does the regional center of the model correspond to the location of the receiver?

line 134: it is not clear why 9 VTEC models are applied per day. It should be specified that a VTEC model is generated over 3 hours of time.

line 166: I suggest to indicate explicitly that the 40/L corresponds to a period of 100 days.

line 178: why only stations providing P1 code measurements of pseudorange were used? Will the result be significantly different if any station would have been selected regardless of the measured code?

On figure 2 an anomaly appears at the end of 2010, where a gap (or values outside the vertical axis limit ?) appears on the estimated shell heights. In this article there is not a discussion about this strange behaviour, but in the previous article (Zhao and Zhou 2018), figure 3 shows that all stations have simultaneously anomalous DCBs during a few months. I suggest to make a deeper investigation on why this happen, but clearly these DCBs values are not reliable. Some hypotheses: an error in CODE processing chains; an error in the receivers firmware that affect the time estimate; some error at GPS system level. . . The impact to the results of this article concern the Fourier model to represent the whole solar cycle behaviour of the shell height, but should not affect the station comparisons of 2014.

Additional comments on Figure 2: -the spreading of daily shell height values is extremely large (>200 km) with strong variations from one day to the other. How this spreading is affected by the choice of elevation mask angle? -If there is such a high

variability, what is the benefit of using a Fourier model up to order 40? A much lower order could provide comparable results. On the other hand, the fast variability is not achievable with this model. -both stations show the limits of the proposed approach: the distributions on the right panels present each a missing tail, suggesting that the imposed shell height limits are not adequate. For GOLD station we could expect shell heights higher than 1000 km and for PTBB shell heights lower than 100 km, which are unphysical, because outside the ionosphere.

Figure 4 and 5 top panels show the difference of the DCBs of 2014 in the reference station with the predictions of the Fourier model. However this model has been presented earlier only in term of shell height. It is therefore difficult to understand if it is a good prediction or not. I think a more explicit discussion of the whole validation approach is needed.

**3 Technical corrections**

line 108: I think "it is intuitional and practical" should read "it is intuitive and practical"

line 191: correct "the receiver type of GOLD have been changed" into "the receiver type of GOLD has been changed".

line 197: I suggest to indicate in the caption that the stations in black are the reference stations for the study. I would also suggest to include in both maps of figure 1 circles centered on the reference station to indicate the distances, e.g. 300, 600, 900, 1200 km, or whichever choice the authors think is significant.

line 201: I suggest to indicate more explicitly that in the table the column of "Receiver type" includes the date of change of the receivers in the reference stations.

Figure 2 vertical axis label contains a typo: Scarge instead of Scargle
lines 239-241: the description of figure 4 repeats the concept expressed in the previous sentence. To avoid confusion I suggest to simplify the text writing something like: "The results of this comparison are shown in Figure 4".

line 252: I suggest to rewrite the sentence "Note that some days no result because of missing data", for instance: "Data gap on the figure correspond to days when data from that station are not available".

lines 254-256: I suggest to simplify the sentence to avoid the cumbersome expression "is more concentrated distributed around 0 in a statistical sense".

lines 256-258 the wording "can be improved" at that position in this sentence is not grammatically correct.

line 320: correct "GLOD" into "GOLD".

Many bibliographic records appear to be incomplete, either the title of the article or the volume number, or doi is missing. Doi should be included without the "https://doi/org/" prefix.

---

## Author Comment (AC2) · 3 Oct 2018

We thank referee #2 for careful reading and valuable comments on the manuscript. Accordingly, we have modified the text and figures. All the modifications and changes are shown in the revised manuscript "manuscript-version3" in red font. Our responses to the referee's comments are listed in the file "Response to RC2". The two files are in supplement.

According your constructive suggestion, we have discussed the limitations of this approach more deeper, and revised the mistakes in the text and figures. The format of reference also has been adjusted according the requirement of ANGEO.

[Figure]

Thank you again for your comments, which help us improve this manuscript significantly.

Please also note the supplement to this comment:
https://www.ann-geophys-discuss.net/angeo-2018-73/angeo-2018-73-AC2-supplement.zip

---

## Referee Report (RR1)

**Review of manuscript "Validation and application of optimal ionospheric shell height model for single-site TEC estimation" by Jiaqi Zhao and Chen Zhou**

This manuscript presents a method to determine the optimal ionospheric shell height (= effective height of an assumed thin ionospheric shell) based on TEC measurements and DCB code biases. The method is a further development and validation of a method developed by the same authors.

In my opinion, this is a useful contribution to the scientific community, particularly since in many applications it has become common practice to "just assume" a fixed ionospheric height (of often about 350 km) without thinking. As criticism, one might argue that physical inputs, such as ionosondes, radar measurements, or solar activity data, have not been used in this method. However, I think that the approach of this paper can be considered just one approach, and is useful to be compared to other approaches which may use other information. Besides, the optimal method to determine the effective ionospheric height may depend on the application, which makes it useful to try different methods.

The paper is mostly clear and well written. The authors have clearly well taken into account the comments made by the two earlier reviewers. I have only a few more questions, see here below. Furthermore, I have made editorial comments to improve the English in the annotated manuscript, onward from page 3 of this pdf-file.

**Comments:**

Equation (1):
Shouldn't the equation also include a term "$T_V(\varphi_0, S_0)$" ? Otherwise, the equation seems to say that VTEC=0 at the regional center.
Or is $T_V(\varphi, S)$ supposed to mean: the difference in VTEC between a station and the regional center?

Equation (5):
- You are applying this optimization over 11 years, and therefore only for the two reference stations, right?
- Does the matrix $\mathbf{\Phi}$ contain the function $f(z)$ from equation (3)?
- And presumably, $\mathbf{\theta}$ contains only 1s and 0s, right?
- What does the matrix $\mathbf{E}$ contain? It cannot be the model coefficients from equation (1), because you are applying this only for the reference stations, so $\varphi - \varphi_0$ and $S - S_0$ are 0. So is $\mathbf{E}$ the vector of $T_V(\varphi, S)$ values as in equation (2)? (= VTEC, in the reference stations)
- If so, where do you get these VTEC values from? From equation (2) using your optimal shell height model?

Please clarify these things in the text.

Line 245-248:
"… the DCBs in all stations of each region are estimated in the form of single station by the polynomial model mentioned earlier."
How do you estimate these?

- Do you mean with the "polynomial model": equation (1)? If so, how do you find the coefficients?
- Or do you mean that you assume that the optimal ionospheric height at the stations is equal to that at the reference station? If so, what do you mean with the "polynomial model"? And when do you use equation (1)?

Line 348-350:
"For different region, the error at 0 km (i.e. the error for the reference station) is different, which should be also considered."
This is a little vague; can you explain more what you mean to say with this? For instance, something like: "The quality of the DCB estimations depends also on the quality of the optimal shell height model at the reference stations themselves, which may also not be equally good in all areas."

**Editorial comments:**

Please see the comments in red, annotated in the copy of the manuscript, below.

[revised manuscript text omitted]

---

## Editor Decision (ED1)

Dear Authors,

I appreciate your attention you paid to the referee´s comments and the efforts you made to improve the manuscript. We have sent the improved manuscript for the second revision, and now I am coming back to you on the status of your paper. The referee read very carefully the improved manuscript and the review we received is quite positive. Nevertheless, the referee requested answers to some questions; also some additional corrections should be made. For your convenience the referee's comments are enclosed below. Please, consider these additional comments and after the minor revision I'll recommend the manuscript to be published.

If you are prepared to undertake the additional work required, please submit a list of changes or a rebuttal against each point, which is being raised when you submit the revised manuscript.

Kindest regards

Yours sincerely

D. Buresova

**Review of manuscript "Validation and application of optimal ionospheric shell height model for single-site TEC estimation" by Jiaqi Zhao and Chen Zhou**

This manuscript presents a method to determine the optimal ionospheric shell height (= effective height of an assumed thin ionospheric shell) based on TEC measurements and DCB code biases. The method is a further development and validation of a method developed by the same authors.

In my opinion, this is a useful contribution to the scientific community, particularly since in many applications it has become common practice to "just assume" a fixed ionospheric height (of often about 350 km) without thinking. As criticism, one might argue that physical inputs, such as ionosondes, radar measurements, or solar activity data, have not been used in this method. However, I think that the approach of this paper can be considered just one approach, and is useful to be compared to other approaches which may use other information. Besides, the optimal method to determine the effective ionospheric height may depend on the application, which makes it useful to try different methods.

The paper is mostly clear and well written. The authors have clearly well taken into account the comments made by the two earlier reviewers. I have only a few more questions, see here below. Furthermore, I have made editorial comments to improve the English in the annotated manuscript, onward from page 3 of this pdf-file.

**Comments:**

Equation (1):
Shouldn't the equation also include a term "$T_V(\varphi_0, S_0)$" ? Otherwise, the equation seems to say that VTEC=0 at the regional center.
Or is $T_V(\varphi, S)$ supposed to mean: the difference in VTEC between a station and the regional center?

Equation (5):
- You are applying this optimization over 11 years, and therefore only for the two reference stations, right?
- Does the matrix $\boldsymbol{\Phi}$ contain the function $f(z)$ from equation (3)?
- And presumably, $\boldsymbol{\theta}$ contains only 1s and 0s, right?
- What does the matrix $\mathbf{E}$ contain? It cannot be the model coefficients from equation (1), because you are applying this only for the reference stations, so $\varphi - \varphi_0$ and $S - S_0$ are 0. So is $\mathbf{E}$ the vector of $T_V(\varphi, S)$ values as in equation (2)? (= VTEC, in the reference stations)
- If so, where do you get these VTEC values from? From equation (2) using your optimal shell height model?

Please clarify these things in the text.

Line 245-248:
"… the DCBs in all stations of each region are estimated in the form of single station by the polynomial model mentioned earlier."
How do you estimate these?

- Do you mean with the "polynomial model": equation (1)? If so, how do you find the coefficients?
- Or do you mean that you assume that the optimal ionospheric height at the stations is equal to that at the reference station? If so, what do you mean with the "polynomial model"? And when do you use equation (1)?

Line 348-350:
"For different region, the error at 0 km (i.e. the error for the reference station) is different, which should be also considered."
This is a little vague; can you explain more what you mean to say with this? For instance, something like: "The quality of the DCB estimations depends also on the quality of the optimal shell height model at the reference stations themselves, which may also not be equally good in all areas."

**Editorial comments:**

Please see the comments in red, annotated in the copy of the manuscript, below.

[revised manuscript text omitted]

---

## Author Response (AR2)

Comments:

The manuscript is well written and clear. This paper models the optimal thin layer altitude as 40th order Fourier series. The optimal altitude coefficients are estimated at a reference station and used for estimating the thin shell altitude of the near stations. This approach is very useful for precisely estimating the TEC and DCB in not IGS platform.

Reply:

We thank the referee for the encouraging evaluation on our study, which has driven us into a deeper investigation.

It would be interesting to see what are the performances in DCB estimation if the number of coefficients of the Fourier series changes, it would be interesting to see what is the minim number of coefficients required.

Reply:

We thank the referee for this valuable comment. In our study, the order of Fourier

series is preliminarily set to 40. For one station, the outstanding frequencies of optimal thin layer altitude are only a few, it is possible to reduce the number of coefficients. Maybe less coefficient, clearer physical relationship.

There is in addition a typo comment, on page 7 line 132, you should insert the IPP acronym that has not been specified before.

Reply:

We thanks the referee for careful reading and pointing out this mistake. We have corrected it. Please see page 7 line 135 in the revised manuscript.

Dear referee #2,

We are grateful for your careful reading and valuable comments on the manuscript. Accordingly, we have modified the text. All the modifications and changes are shown in the revised manuscript in red font. Our responses are listed below.

**General comments**

This article presents a novel approach to estimate GPS permanent stations DCBs in a radius less than 2000 km from a mid-latitude IGS station. It is a validation of a technique recently developed by the same authors. An optimum ionospheric shell height is estimated using the assumption that the IGS DCBs represent reliable values. This study covers a complete solar cycle for the estimation of the ionospheric shell height at a reference station and one year for the tests with additional stations. I think that the manuscript in its present form lacks of necessary discussion on the limitations of the assumptions made in this work and that a number of points need to be explained deeper. I therefore suggest major revisions.

**Specific comments**

The ionospheric height for the reference station GOLD appear to be very high: in average it is 712 km. By an ionospheric point of view, the shell height should correspond to the height of the ionosphere barycenter, i.e. higher than hmF2 of about 100-150 km. There is a long-lasting debate on the operational shell height to use for the thin shell approximation of the ionosphere and the authors recall many of the publications discussing this problem. While it is true that some authors allow altitudes as high as 1200 km, care should be taken to understand if the obtained shell height are reliable. In this work an elevation mask of 15° has been used. Under many conditions this elevation mask could be too low and introduce a large uncertainty on the optimum shell height (see for instance the discussions of Rama Rao et al. 2006,

recall that under some conditions they even obtained unphysical negative shell height).

Reply:

We thank for your important remark. We agree with that the shell height should correspond to the height of the ionosphere barycentre by an ionospheric point of view. While for accurate TEC and DCB estimation, because of VTEC model error and mapping function error and so on, optimal shell height is different with ionosphere barycentre. Actually for different VTEC model, the optimal shell height is also different. Lu et al. (2017) did the similar work by using another ionospheric shell height estimation method. In our manuscript, the optimal shell height is also affected by the accuracy of reference values of DCB. The optimal shell height is more like a modification of the mapping function for the selected VTEC model, and have relationship with solar activity. We believe that optimal shell height and ionosphere barycentre could be closer with the improvement of the VTEC model and mapping function.

Reference

Lu W, Ma G, Wang X, Wan Q, Li J (2017) Evaluation of ionospheric height assumption

for single station GPS-TEC derivation. Advances in Space Research 60(2):286-294

It is not clear why the technique proposed for ionospheric shell height estimation cannot be implemented to isolated GNSS receivers not belonging to IGS stations (line 107).

Reply:

We thank for your carefully reading and helpful comment. The optimal ionospheric shell height is calculated from IGS DCB values. DCB is normally not released by non-IGS stations, which means ionospheric shell height cannot be calculated by using this method. However, if we could get the long-term observations and reference values of DCB from non-IGS station, this technique could also work. We have deleted

this mistake. Please see page 6 line 107-109 in the revised manuscript.

A Fourier model of the shell height is constructed for GOLD and PTBB for a complete solar cycle between 2003 and 2013. This model does not include any input regarding solar activity. It is well known that the current solar cycle is considerably less strong than the previous one. The ionosphere development has also been substantially lower. Thus it is also expected the optimum shell height should follow a different pattern. A discussion on this point is essential for the correct understanding of this work.

Reply:

We thank for your constructive suggestion. We total agree with the reviewer that solar activity is the dominant factor for ionospheric variability. However, other factors such as atmospheric variability and human activity can also cause ionospheric disturbance. In this study, we do not consider all physical factors explicitly. However, we try to include all the factors by utilizing empirical modeling with data. The Fourier model is a preliminary result. Evaluations on different models will be investigated and compared in the following work.

line 52: the work of Sardón et al. (1994) was not oriented towards real-time ionospheric VTEC, but to develop a technique of prediction of DCBs under adverse conditions (antispoofing, ionospheric disturbances).

Reply:

We appreciate the reviewer for this helpful comment. We have accordingly made the revision. Please see Lines 52-54 in the revised manuscript.

line 77: specify that the Nava et al. (2007) technique uses multiple stations to obtain a "coinciding pierce point".

Reply:

We thank for the reviewer for providing this suggestion. We have accordingly made the revision. Please see Lines 77 in the revised manuscript.

line 125-126: the polynomial model is referred to Lanyi and Roth (1988). However the expression used in this article does not correspond to the one used by those authors.

Reply:

We appreciate the reviewer for pointing out this mistake. We have replaced this reference with (Wild, 1994; Komjathy, 1997). Please see Lines 128-129 in the revised manuscript.

Line 132: does the regional center of the model correspond to the location of the receiver?

Reply:

Yes. The regional center of the model is the location of the receiver.

line 134: it is not clear why 9 VTEC models are applied per day. It should be specified that a VTEC model is generated over 3 hours of time.

Reply:

We have accordingly made the revision. Please see Lines 137-139 in the revised manuscript.

line 166: I suggest to indicate explicitly that the 40/L corresponds to a period of 100 days.

Reply:

We thank for this helpful suggestion. We have accordingly write it explicitly. Please see Lines 170-171 in the revised manuscript.

line 178: why only stations providing P1 code measurements of pseudorange were used? Will the result be significantly different if any station would have been selected regardless of the measured code?

Reply:

We thank for this comment. CODE also provides the DCB of P1-C1, but only for

satellites. And we are not sure whether the receiver DCB is C1-P2 bias in CODE DCB file, for the station providing C1 code but no P1 code. So we use the DCB of P1-P2 for reference. Accordingly, we just select stations with P1 code.

On figure 2 an anomaly appears at the end of 2010, where a gap (or values outside the vertical axis limit ?) appears on the estimated shell heights. In this article there is not a discussion about this strange behavior, but in the previous article (Zhao and Zhou 2018), figure 3 shows that all stations have simultaneously anomalous DCBs during a few months. I suggest to make a deeper investigation on why this happen, but clearly these DCBs values are not reliable. Some hypotheses: an error in CODE processing chains; an error in the receivers firmware that affect the time estimate; some error at GPS system level. The impact to the results of this article concern the Fourier model to represent the whole solar cycle behavior of the shell height, but should not affect the station comparisons of 2014.

Reply:

We appreciate the reviewer for raising this important comment. We totally agree with the reviewer that the anomaly at the end of 2010 could be an error in CODE processing procedures. We have followed the reviewer's suggestion that discussion on the data gap have been added in the revised manuscript. Please see Line 217-219.

[Figure]

**Fig.1** Receiver's DCB released by CODE, IGS and JPL from 2010 to 2011

Figure 1 plots the DCB of receiver provided by different analysis center. At the end of 2010, for CODE, PTBB and GOLD both appear anomaly; for IGS, only PTBB is anomalous; the DCB of GOLD provided by JPL seems continuous. JPL doesn't release the DCB of PTBB after April 2010. It seems that the DCB provided by JPL is more reliable, in terms of stability.

Additional comments on Figure 2: -the spreading of daily shell height values is extremely large (>200 km) with strong variations from one day to the other. How this spreading is affected by the choice of elevation mask angle? -If there is such a high variability, what is the benefit of using a Fourier model up to order 40? A much lower order could provide comparable results. On the other hand, the fast variability is not achievable with this model. -both stations show the limits of the proposed approach: the distributions on the right panels present each a missing tail, suggesting that the imposed shell height limits are not adequate. For GOLD station we could expect shell heights higher than 1000 km and for PTBB shell heights lower than 100 km, which are unphysical, because outside the ionosphere

Reply:

We thank for this constructive suggestion. We set the elevation cut-off angle as 15° and 30°,Figure 2 shows their results. When elevation mask angle is set as 30°, the spreading is larger, compare to 15°. But their optimal ionospheric shell heights have similar fluctuation frequencies. When other stations around apply the model, their elevation mask angles must to be same with the reference station.

[Figure]

**Fig.2** the optimal ionospheric shell height with different elevation mask angle at PTBB

Figure 3 and Figure 4 show the Fourier models with different order from 10 to 40 for GOLD and their errors. With the increase of order, more details display, the variance of fitting error decreases. The models with 35 order and 40 order are similar, and much different with the other orders. Figure 3 (in the manuscript) shows that the 4-month cycle is also outstanding at GOLD. If we set the order as 30 (the minimum cycle is about 134 days) or smaller, the 4-month cycle will lost. So we conservatively set the order as 40.

[Figure]

**Fig.3** Fourier fitting results of different order for GOLD

[Figure]

**Fig.4** Errors of the Fourier fittings for GOLD and their Gaussian fitting results

Yes, the missing tails indicate the limits of our approach. The approach attempts to reduce the DCB error by modifying the shell height. While the error is not only caused by the inappropriate shell height, but also caused by mapping function error and VTEC model error.

Figure 4 and 5 top panels show the difference of the DCBs of 2014 in the reference station with the predictions of the Fourier model. However this model has been presented earlier only in term of shell height. It is therefore difficult to understand if it is a good prediction or not. I think a more explicit discussion of the whole validation approach is needed

Reply:

We appreciate the reviewer for raising this comment. We have followed the suggestion. Please see Line 172-176. The difference between DCB released by CODE and DCB calculated using the predicted optimal ionospheric shell heights are plotted in red dots. The difference between DCB released by CODE and DCB calculated using the fixed ionospheric shell height are plotted in black dots. In both Figure 4 and Figure 5, the general distribution and mean value of red dots is smaller than that of black dots, which means the DCB estimation is improved by using the predicted

optimal ionospheric shell heights.

**Technical corrections**

line 108: I think "it is intuitional and practical" should read "it is intuitive and practical"

Reply:

We thank the reviewer for pointing out this mistake. We have accordingly make the revision. Please see Line 112 in the revised manuscript.

line 191: correct "the receiver type of GOLD have been changed" into "the receiver type of GOLD has been changed".

Reply:

We thank the reviewer for pointing out this mistake. We have accordingly make the revision. Please see Line 198 in the revised manuscript.

line 197: I suggest to indicate in the caption that the stations in black are the reference stations for the study. I would also suggest to include in both maps of figure 1 circles centered on the reference station to indicate the distances, e.g. 300, 600, 900, 1200 km, or whichever choice the authors think is significant.

Reply:

We thank for your helpful comment. We have modified figure 1 as suggested. Please see Line 203 in the revised manuscript.

line 201: I suggest to indicate more explicitly that in the table the column of "Receiver type" includes the date of change of the receivers in the reference stations.

Reply:

We thank for this comment. We have followed the suggestion. Please see Line 208 in the revised manuscript.

Figure 2 vertical axis label contains a typo: Scarge instead of Scargle

Reply:

We thank the reviewer for pointing out this mistake. We have modified figure 3 as suggested. Please see Line 235 in the revised manuscript.

lines 239-241: the description of figure 4 repeats the concept expressed in the previous sentence. To avoid confusion I suggest to simplify the text writing something like: "The results of this comparison are shown in Figure 4".

Reply:

We thank the reviewer for pointing out this mistake. We have accordingly make the revision. Please see Line 249 in the revised manuscript.

line 252: I suggest to rewrite the sentence "Note that some days no result because of missing data", for instance: "Data gap on the figure correspond to days when data from that station are not available".

Reply:

We thank the reviewer for pointing out this mistake. We have accordingly make the revision. Please see Line 259-260 in the revised manuscript.

lines 254-256: I suggest to simplify the sentence to avoid the cumbersome expression "is more concentrated distributed around 0 in a statistical sense".

Reply:

We thank for the reviewer for raising this suggestion. We have followed the reviewer's suggestion and rephrased this sentence. Please see Lines 261-264 in the revised manuscript.

lines 256-258 the wording "can be improved" at that position in this sentence is not grammatically correct.

Reply:

We thank for reviewer for this important comment. We have rephrased this sentence

in the revised manuscript. Please see line 264-266 in the revised manuscript.

line 320: correct "GLOD" into "GOLD".

Reply:

We thank the reviewer for pointing out this mistake. We have accordingly make the revision. Please see Line 329 in the revised manuscript.

Many bibliographic records appear to be incomplete, either the title of the article or the volume number, or doi is missing. Doi should be included without the "https://doi/org/" prefix.

Reply:

We thank the reviewer for the helpful suggestions. We have revised the references accordingly in the revised manuscript.

**Response to Referee #3**

Dear referee #3,

We appreciate your careful reading and valuable review.

A detailed explanation and justification for your comments are listed below in this page.

According to the editorial comments, we have modified the text. All the modifications and changes are shown in the revised manuscript in red font from page.

Thanks for your reading and comments again.

Kindest regards

Yours sincerely

Jiaqi Zhao

**COMMENTS TO THE AUTHOR:**

**General comments**

This manuscript presents a method to determine the optimal ionospheric shell height (= effective height of an assumed thin ionospheric shell) based on TEC measurements and DCB code biases. The method is a further development and validation of a method developed by the same authors.

In my opinion, this is a useful contribution to the scientific community, particularly since in many applications it has become common practice to "just assume" a fixed ionospheric height (of often about 350 km) without thinking. As criticism, one might argue that physical inputs, such as ionosondes, radar measurements, or solar activity data, have not been used in this method. However, I think that the approach of this paper can be considered just one approach, and is useful to be compared to other approaches which may use other information. Besides, the optimal method to

determine the effective ionospheric height may depend on the application, which makes it useful to try different methods.

The paper is mostly clear and well written. The authors have clearly well taken into account the comments made by the two earlier reviewers. I have only a few more questions, see here below. Furthermore, I have made editorial comments to improve the English in the annotated manuscript, onward from page 3 of this pdf-file.

Replay:

We thank the referee for the encouraging evaluation on our study.

**Specific comments**

Equation (1): Shouldn't the equation also include a term "$T_V(\varphi_0, S_0)$" ? Otherwise, the equation seems to say that VTEC=0 at the regional center. Or is $T_V(\varphi, S)$ supposed to mean: the difference in VTEC between a station and the regional center?

Reply:

We thank the referee for this comment.

The term "$T_V(\varphi_0, S_0)$" is included in equation (1). According to equation (1),

$$T_V(\varphi_0, S_0) = \sum_{i=0}^{m} \sum_{j=0}^{n} E_{ij} (\varphi_0 - \varphi_0)^i (S_0 - S_0)^j = E_{00}.$$

Equation (1): $\quad T_V(\varphi, S) = \sum_{i=0}^{m} \sum_{j=0}^{n} E_{ij} (\varphi - \varphi_0)^i (S - S_0)^j$

This VTEC model fits the VTEC over a period of time, time is an input parameter. In the VTEC model, $S = 15 \cdot t + \gamma$, where $t$ is UTC time (hour), $\gamma$ is IPP longitude (degree).

Equation (5):

• You are applying this optimization over 11 years, and therefore only for the two reference stations, right?

Reply:

Yes, Equation (5) are applied to estimate the optimal ionospheic shell heights over 11

years, based on the data of the two reference stations. Then two models are established by fitting the estimated optimal ionospheic shell heights. And other stations are used to verify whether the two models can be applied to nearby stations.

Equation (5): $\min\limits_{100<h<1000} mean\left(\left|\mathbf{DCB_{ref}}-\mathbf{DCB}\right|\right)$ s.t. $\mathbf{T}=\boldsymbol{\Phi}\cdot\mathbf{E}+\boldsymbol{\theta}\cdot\mathbf{DCB}$

• Does the matrix **ф** contains the function *f(z)* from equation (3)?

Reply:

We thank the referee for this comment.

The matrix **ф** contains $(\varphi-\varphi_0)^i(S-S_0)^j f(z)$.

Please see Line 167-168 in the revised manuscript.

• And presumably, **θ** contains only 1s and 0s, right?

Reply:

Yes, for example, the i-th STEC corresponds to the j-th satellite, then **θ**(i,j)=1, and **θ**(i,k)=0, k≠j.

Please see Line 168-169 in the revised manuscript.

• What does the matrix **E** contain? It cannot be the model coefficients from equation (1), because you are applying this only for the reference stations, so $\varphi-\varphi_0$ and $S-S_0$ are 0. So is **E** the vector of $T_V(\varphi,S)$ values as in equation (2)? (= VTEC, in the reference stations)

Reply:

We thank the referee for this valuable comment. It's our mistake. We do not make it clear. In the two cases, all of the DCB estimations are based on single site.

Only the two reference stations are applied to establish two optimal shell height models using equation (1)-(6). Each station is applied to estimate DCB separately in 2014 using equation (1)-(4), same as the general single site DCB estimation (one station, one VTEC model), except the shell heights are provided by the optimal shell height models. Their VTEC models (i.e. $T_V(\varphi,S)$) are independent of each other, and

the VTEC model coefficients ($E_{ij}$) are unknown.

So the matrix **E** contains $E_{ij}$.

"$\varphi_0$ and $S_0$ denote $\varphi$ and $S$ at regional center" (line 137-138 in the manuscripts), "the region" here means the cover region of IPP for single station in one day, not the regions in figure 1. So "regional center" means the center of the cover region of IPP. And for different station, $\varphi_0$ and $S_0$ are different. $\varphi$ and $S$ are the location of IPP, they change all the time.

For example, the steps of estimating the DCB of TABV in 2014-1-1:

Frist, the optimal shell height model based on GOLD (the reference station near TABV) provides the predicted shell height in 2014-1-1, and $h$ in equation (3) is set as the predicted shell height.

Then the data of TABV in 2014-1-1 and $h$ are substituted into equation (1)-(4). The data of TABV contains raw STEC ($T_{OS}^{PRN}$), IPP location ($\varphi,S$), the center of IPP ($\varphi_0,S_0$, you can simplify set them as the location of TABV), and elevation angel ($El$). So only the VTEC model coefficients ($E_{ij}$) and DCB are unknown.

Finally, the VTEC model coefficients ($E_{ij}$) and DCB are estimated by least square method. Equation (1)-(4) can be written as:

$$\mathbf{T} = \mathbf{\Phi} \cdot \mathbf{E} + \mathbf{\theta} \cdot \mathbf{DCB} = \begin{bmatrix} \mathbf{\Phi} & \mathbf{\theta} \end{bmatrix} \cdot \begin{bmatrix} \mathbf{E} \\ \mathbf{DCB} \end{bmatrix}$$

We have modified, please see Line 137-138 in the revised manuscript.

• If so, where do you get these VTEC values from? From equation (2) using your optimal shell height model?

Reply:

We thank the referee for this comment. Each station has its own VTEC model, the VTEC model coefficients are unknown, and are estimated with DCB.

Please clarify these things in the text.

Reply:

We thank the referee for this comment. We have modified. Please see line 137-138 and line 167-169 in the revised manuscript.

Line 245-248: "… the DCBs in all stations of each region are estimated in the form of single station by the polynomial model mentioned earlier." How do you estimate these?

Reply:

We thank the referee for this valuable comment. The DCBs are estimated by Equation (1)-(4), and can be written as:

$$\mathbf{T} = \mathbf{\Phi} \cdot \mathbf{E} + \mathbf{\theta} \cdot \mathbf{DCB} = \begin{bmatrix} \mathbf{\Phi} & \mathbf{\theta} \end{bmatrix} \cdot \begin{bmatrix} \mathbf{E} \\ \mathbf{DCB} \end{bmatrix}$$

where, E and DCB is unknown, and can be estimated by least square method.

"the polynomial model mentioned earlier" is ambiguity, it means the polynomial VTEC model (equation (1)). We have replaced it with "equation (1)-(4)", please see line 250-251 in the revised manuscript.

• Do you mean with the "polynomial model": equation (1)? If so, how do you find the coefficients?

Reply:

Yes, "polynomial model" is expressed as equation (1), the coefficients are unknown, and are estimated with DCB. See the reply above.

• Or do you mean that you assume that the optimal ionospheric height at the stations is equal to that at the reference station? If so, what do you mean with the "polynomial model"? And when do you use equation (1)?

Reply:

Yes, the two optimal ionospheric shell height models provide shell height to the stations nearby. "polynomial model" means the polynomial VTEC model which is expressed as equation (1). Equation (1) is used in DCB estimation and the optimal

ionospheric shell height estimation.

Line 348-350: "For different region, the error at 0 km (i.e. the error for the reference station) is different, which should be also considered." This is a little vague; can you explain more what you mean to say with this? For instance, something like: "The quality of the DCB estimations depends also on the quality of the optimal shell height model at the reference stations themselves, which may also not be equally good in all areas."

Reply:

We thank the referee for this helpful comment.

Yes, the quality of the DCB estimations also depends on the quality of the optimal shell height model at the reference stations themselves. Figure 6 and Figure 7 show the error of DCB by the optimal ionospheric shell height increases linearly with the distance to the reference station GOLD or PTBB. So the quality of the DCB estimations depends on the slope, the distance and the error at 0 km.

We have modified this part, please see line 351-353 in the manuscripts.

**Editorial comments**

Please see the comments in red, annotated in the copy of the manuscript, below.

Reply:

We thank the referee for this helpful comment and carefully reading. The editorial comments is very helpful to improve our manuscripts. Thank you very much! We have accordingly make the revision.

[revised manuscript text omitted]